MicroRNA-21 as a potential biomarker for detecting esophageal carcinoma in Asian populations: a meta-analysis

http://orcid.org/0000-0002-5394-6313 Han Zheng
Pan Lingbo
Lu Bangjie
Zhu Huixia hanzhengzhx@163.com
Department of Clinical Medicine, Medical School of Nantong University, Nantong University , Nantong , China
Falfán-Valencia Ramcés
Electronic publication date: 2022 Sep 30
Publication date: 2022
Volume: 10
Electronic Location ID: e14048
Received 2022 Jun 7; Accepted 2022 Aug 22
Copyright: © 2022 Han et al.
Copyright year: 2022
Copyright holder: Han et al.
License: This is an open access article distributed under the terms of the Creative Commons Attribution License, which permits unrestricted use, distribution, reproduction and adaptation in any medium and for any purpose provided that it is properly attributed. For attribution, the original author(s), title, publication source (PeerJ) and either DOI or URL of the article must be cited.
License URL: https://creativecommons.org/licenses/by/4.0/

Keywords: MicroRNA-21, Esophageal carcinoma, Diagnosis, Meta-analysis, Asian

Funding: The authors received no funding for this work.

==============================
Background

MicroRNA-21 (miR-21) is significantly expressed in a variety of cancers and could be used as a tumor biomarker. However, the results are varied, and no studies on the diagnostic usefulness of miR-21 in Asian esophageal cancer (EC) patients have been published. This meta-analysis was aimed at exploring whether miR-21 can be used as a diagnostic marker and assessing its effectiveness.

Methods

The relevant literature was identified in six main databases: Ovid MEDLINE, PsycINFO, PubMed MEDLINE, Embase, Web of Science, and the Cochrane Library. Two researchers independently selected the literature based on the inclusion and exclusion criteria, extracted data, and evaluated the risk of bias. The meta-analysis was carried out using Review Manager 5.4, Meta-Disc 1.4 and STATA 15.1 software. In the end, 987 patients from 12 different studies were included. Quality evaluation of diagnostic accuracy studies 2 (QUADAS—2) was used to examine the risk of bias.

Results

The pooled sensitivity (SEN) was 0.72 (95% CI [0.69–0.75]), the pooled specificity (SPE) was 0.78 (95% CI [0.75–0.81]), the pooled positive likelihood ratio (PLR) was 2.87 (95% CI [2.28–3.59]), the pooled negative likelihood ratio (NLR) was 0.36 (95% CI [0.31–0.43]), the pooled diagnostic odds ratio (DOR) was 10.00 (95% CI [7.73–12.95]), and the area under the curve 0.82 (95% CI [0.79–0.85]). A Deeks’ funnel plot shows that there was no publication bias (P = 0.99).

Conclusion

Our findings suggest miR-21 might be the potential biomarker for detecting EC in Asian populations, with a good diagnostic value.

Introduction

Esophageal carcinoma (EC) is the seventh most frequent cancer in terms of incidence, affecting more than 450,000 people globally (Li et al., 2020). The majority of esophageal cancers can be subdivided into two main histological subtypes: adenocarcinomas (AC) and squamous cell carcinomas (SCC). Esophageal AC incidence has mysteriously increased in white people in high-income nations in recent years. This increase has largely been attributed to the rising prevalence of obesity, although some controversy exists around this hypothesis (Arnold et al., 2015). SCC accounts for about 88 percent of esophageal cancer cases all over the world, particularly in Central and South-eastern Asia. In 2017, China accounted for nearly half of all new cases of EC (234,624 Chinese cases out of 472,525 cases worldwide, or 49.7%) (Li et al., 2021). Nitrosamine compounds, a class of the most probable carcinogens concerned with EC, are found more frequently and at a higher concentration in Asian foods than in Western foods (Jakszyn & Gonzalez, 2006; Zhao et al., 2020; Stoner, Wang & Chen, 2007). A recent comprehensive analysis predicts that the global burden of EC will continue to rise (Ma et al., 2020).

EC is a devastating disease, with a 5-year survival rate of fewer than 20% and the highest mortality rate in Asia (Yu, Tan & Liu, 2019). However, if only the mucosal layer of the esophagus or the superficial submucosal layer is invaded, the 5-year survival rate can be as high as 95%. Unfortunately, the majority of EC patients are diagnosed at an advanced stage of the disease (Guo et al., 2016). As a result, early detection of EC can dramatically improve the prognosis and decrease mortality. Biopsy and endoscopy, the current gold standard procedures for detecting and diagnosing EC, are intrusive and accurate diagnosis can be attained by biopsy only when patient have reached advanced stage (Fan et al., 2019; Wang et al., 2014; Liu et al., 2016; Zhao, 2012). Meanwhile, endoscopic and pathological doctors require a lengthy qualification process (Zhao, 2012). A number of biomarkers reflecting the molecular status or activity within the tumors have been used for the clinical diagnosis of cancers (Xie et al., 2022; Qiu et al., 2021; Qiu et al., 2022). Conventional tumor markers, such as carcinoembryonic antigen (CEA), carbohydrate antigen (CA) 19-9, and squamous cell carcinoma antigen (SCC), have been used in diagnostic assays for early detection and monitoring the tumor dynamics of EC. These tumor markers, however, lack sufficient sensitivity and specificity to facilitate early detection of EC (Zarrilli et al., 2021).

MicroRNA-21 (miR-21) is found in a range of extracellular fluids, including cerebrospinal fluid (CSF), saliva, serum, plasma, and blood, and has been reported to be highly expressed in a variety of cancer types (Bautista-Sánchez et al., 2020). MiR-21 has several isoforms, the most important, miR-21-3p and miR-21-5p. Gao et al. (2019) reported that up-regulation of hsa-miR-21-3p was associated with a high risk of EC. Meanwhile, miR-21-5p, contributing to the polarization of M2 macrophages in tumor micro-environment, was also increased in patients with EC (Song et al., 2021).

A growing number of studies have suggested the feasibility of miRNAs to diagnose EC patients. However, these studies have returned inconsistent results. Liu et al. (2016) reported a good diagnostic characteristic with 79.9% sensitivity, whereas Xie et al. (2013) reported a much lower specificity (51.3%) for miR-451. Wang et al. (2014) conducted a meta-analysis including only eight publications whose publication years ranged from 2010 to 2013, and yielded moderate diagnostic accuracy, with 78.5% sensitivity and 96% specificity. To advance knowledge based on the existing evidence and to address the limitations of the previous reviews, we undertook this systematic analysis to further explore the clinical applicability of miR-21 as novel biomarker for the diagnosis of EC in Asians.

Methods

We conducted and reported this systematic review according to prespecified criteria outlined by the PRISMA guidelines; the corresponding checklists are shown in the Supplemental File. The study protocol was registered with the PROSPERO international prospective register of systematic reviews (number CRD42021289781).

Search strategy

One investigator (HZ), a health information specialist, searched six databases: Ovid MEDLINE, PsycINFO, PubMed MEDLINE, Embase, Web of Science, and the Cochrane Library. There were no specified date, age, sex, or language restrictions. The coverage dates for this study began from each database’s inception (MEDLINE, 1946; PsycINFO, 1806; Embase, 1947; Web of science,1900; and Cochrane Library, 1995) and ended on December 1, 2021. The search strategy contained three core components, which were linked using the AND operator: disease type (e.g., Esophageal Neoplasms, Neoplasm Esophageal); research method (e.g., sensitivity, predictive value of tests); diagnostic method (e.g., MicroRNA, Primary miRNA). The search was developed initially for PubMed MEDLINE and then adapted for each of the other five databases by mapping the search terms to additional controlled vocabulary and subject heading terminology.

Inclusion criteria

Asian people diagnosed with esophageal cancer (AC, SCC or both);

The included article at least reported sensitivity and specificity of miR-21 in diagnosis of EC patients;

Studies providing the diagnostic performance of miR-21 in blood, serum, plasma, tissue for EC including World Health Organization (WHO) grade I–IV gliomas;

Exclusion criteria

Meta-analysis, reviews, letters, comments, and duplicated studies;

Studies that did not estimate the diagnostic role of miR-21 in EC (AC, SCC or both);

Articles that were published repeatedly or had qualitative outcomes;

Data extraction

The first author’s name, publication year, country, specimen, sample size, specificity, sensitivity, number of patients and controls were all obtained from each study’s patients. Any disagreements among the researchers were worked out with the help of a third researcher.

Quality assessment

Quality Assessment of Diagnostic Accuracy Studies 2 (QUADAS-2) was used to appraise the applicability of included studies using Review Manager 5.4 software. The patient selection, index test, reference standard, and flow and timing domains made up the four domains of this scale. With the exception of the flow and timing domain, where the applicability concern did not apply, each signaling question was rated as “yes,” “no,” or “unclear,” and each study’s risk of bias and worry for application were estimated as “high,” “low,” or “unclear,” respectively. A “yes” response suggested that the risk of bias could be regarded as low, whereas a “no” or “unclear” response indicated that the risk of bias might be rated as high.

Q tests and I2 statistics

Q tests and I2 statistics were used to estimate the heterogeneity caused by a non-threshold effect among the included studies. Meta-Disc1.4 software was used to perform the statistical analysis for this meta-analysis. According to the Cochrane handbook, either P < 0.1 or I2 > 50% suggested the existence of substantial heterogeneity, a random-effects model was applied to quantify the pooled sensitivity, specificity, the positive likelihood ratio (PLR), negative likelihood ratio (NLR), summary diagnostic odds ratio (DOR) and area under curve (AUC). Otherwise, a fixed-effects model was used.

Subgroup analysis

Subgroup analysis was carried out to investigate the potential influential factors on the summary sensitivity and specificity, which included country (China vs Others), specimen types (plasma vs serum vs blood vs saliva), cancer stage (I–IV vs Others), gender (male vs female) and age (>60 vs ≤60). The analyses were performed in StataSE15.1 (Stata Corp LP, College Station, TX, USA) and Meta-Disc1.4.

Sensitivity analysis

To determine whether our analysis was stable, sensitivity analysis was conducted by removing the included studies one by one and analyzing the SROC curve. Sensitivity analysis was conducted with StataSE15.1.

Fagan’s nomogram

Fagan’s nomogram was performed using the “Midas” module in the STATA 15.1 to calculate pre-test probability and post-test probability to assess the diagnostic power of miR-21 in clinical practice.

Publication bias

In addition, we adopted Deeks’ funnel plots to examine the possibility of publication bias across studies, with values showing P < 0.10 considered to have significant statistical publication bias. StataSE15.1 software was used to perform the statistical analysis.

Results

The search identified 532 full texts, of which 520 articles were omitted for the following reasons: (1) reviews, letters or comments; (2) laboratory studies; (3) not directly related to specific outcome; (4) non-digestive system cancer; (5) outcomes were not protein expression. Manual search of references cited in the published studies did not reveal any additional articles. A flow diagram of the search process is shown in Fig. 1.

Figure 1 Study flow diagram.

The meta-analysis includes 12 studies with 14 sets of data involving 987 participants that were published between 2011 and 2019. Table 1 shows the baseline characteristics of the 12 studies.

Table 1 Characteristics of the 12 studies included in this systematic review.

Study	Country	Year	Sample type	Cancer	Control	TNM	SEN (%)	SPE (%)	Median age	Male (P)	Female (P)	Male (C)	Female (C)	AUC	
Sun et al. (2019)	China	2019	Plasma	125	125	I–IV	61.00	90.00	63 (59–68)	76	49	76	49	0.86	
Wang et al. (2019) (a)	China	2019	Plasma	128	45	I	80.00	71.43	N/A	N/A	N/A	N/A	N/A	0.82	
Wang et al. (2019) (b)	China	2019	Plasma	128	45	II	67.02	79.41	N/A	N/A	N/A	N/A	N/A	0.76	
Wang et al. (2019) (c)	China	2019	Plasma	128	45	III	80.85	70.00	N/A	N/A	N/A	N/A	N/A	0.73	
Samiei et al. (2019)	Iran	2019	Plasma	34	34	I–IV	77.78	65.38	50 (48–52)	5	29	5	29	0.70	
Zhang et al. (2018)	China	2018	Blood	125	125	I–IV	74.00	78.00	N/A	N/A	N/A	N/A	N/A	0.80	
Wang et al. (2018)	China	2018	Serum	35	32	I–IV	77.40	82.90	51 (27–83)	26	9	19	13	0.80	
Sharma, Saraya & Sharma, 2018	India	2018	Blood	24	21	I–IV	83.33	57.15	74	18	6	N/A	N/A	0.69	
Komatsu et al. (2016)	Japan	2016	Plasma	37	20	I–IV	54.20	94.30	64	30	7	N/A	N/A	0.68	
Xie et al. (2013)	China	2013	Saliva	39	19	II–IV	89.70	47.40	58 (46–88)	31	8	15	4	0.70	
Xie et al. (2012)	China	2012	Saliva	32	16	I–IV	87.50	62.50	61 (43–88)	26	6	13	3	0.76	
Wang & Zhang (2012)	China	2012	Serum	31	39	I–IV	71.00	69.20	61 (46–82)	23	8	9	30	0.74	
Kurashige et al. (2012)	Japan	2012	Serum	71	39	I–IV	46.50	100.00	70	66	5	N/A	N/A	0.85	
Komatsu et al. (2011)	Japan	2011	Plasma	50	20	I–IV	88.00	70.00	66 (58–74)	44	6	N/A	N/A	0.82	
Note:

TNM, Cancer Stage; SEN, Sensitivity; SPE, Specificity; P, Patient; C, Control; NA, Not Available; AUC, Area under curve. (a), (b), (c) refer to three different data sets from Wang et al. (2019).

Quality assessment

The QUADAS-2 checklist revealed that all of the articles included were of medium to high quality (Fig. 2).

Figure 2 Risk of bias and applicability concerns summary and graph.

Threshold effect and non threshold effect

Spearman correlation coefficient of sensitivity and 1-specificity yielded −0.12 (P = 0.86), indicating no heterogeneity resulting from threshold effect.

Data analysis

The random-effects model was used in this study. We analyzed the combined results of some diagnostic study statistic measures and found a sensitivity of 0.72 (95% CI [0.68–0.75]) (Fig. 3A), specificity of 0.78 (95% CI [0.75–0.81]) (Fig. 3B). The summary receiver operating characteristic curve (SROC) for the included studies showed the area under curve (AUC) was 0.82 (95% CI [0.79–0.85]) (Fig. 4), indicating a good diagnostic accuracy. PLR and NLR have been considered more clinically valuable compared to the specificity and sensitivity. In our meta-analysis, the combined NLR was 0.36 (95% CI [0.31–0.43]) (Fig. 3C) and the combined PLR was 2.87 (95% CI [2.28–3.59]) (Fig. 3D). A PLR value of 2.87 indicated that the probability of a person with a positive test result having EC was about three-fold higher compared to patients without EC. The NLR refers to the probability of a person who has EC testing negative divided by the probability of a person who does not have EC testing negative. Here, in this meta-analysis, we found an NLR value of only 0.36. These results revealed that MiRNA-21 is clearly able to distinguish EC patients from healthy people. The DOR was 10.00 (95% CI [7.73–12.95]) (Fig. 5), suggesting the potential of a 10-fold higher level of miRNA-21 in subjects with positive EC diagnosis compared with subjects with negative results, indicating a high diagnostic accuracy.

Figure 3 Forest plots of (A) sensitivity, (B) specificity, (C) pooled specificity and (D) the pooled sensitivity, of miRNA for diagnosis of EC in Asia.

Figure 4 SROC curve for miR-21 in EC diagnosis.

The numbers within the circles represent their order in Table 1.

Figure 5 Forest plot of included studies assessing the DOR of miR-21 in EC.

Subgroup analysis

As for diagnostic performance, we performed a subgroup analysis on country (China vs Others), specimen types (plasma vs serum vs blood vs saliva), cancer stage (I–IV vs Others), gender (male vs female) and age (>60 vs ≤60). The results revealed that: I–IV stage had better diagnostic values than non-I–IV (DOR, 10.63 vs 8.97); serum had better diagnostic values than other specimen types (DOR, 13.05 > 10.28 > 9.47 > 9.30) (PLR, 4.59 > 3.16 > 2.64 > 1.88) (Figs. 6A–10E); male had better diagnostic values than female (DOR, 10.82 vs 8.97); age >60 had better diagnostic values than age ≤60 (DOR, 11.47 vs 9.10). Table 2 shows the detailed results of subgroup analysis.

Figure 6 Forest plots of (A) sensitivity, (B) specificity, (C) pooled specificity and (D) the pooled sensitivity of plasma miRNA for diagnosis of EC in Asia.

Figure 7 Forest plot of included studies assessing the DOR of plasma miR-21 in EC.

Figure 8 Forest plots of (A) sensitivity, (B) specificity, (C) pooled specificity, (D) the pooled sensitivity and (E) the pooled diagnostic odds ratio of saliva miRNA for diagnosis of EC in Asia.

Figure 9 Forest plots of (A) sensitivity, (B) specificity, (C) pooled specificity, (D) the pooled sensitivity and (E) the pooled diagnostic odds ratio of blood miRNA for diagnosis of EC in Asia.

Figure 10 Forest plots of (A) sensitivity, (B) specificity, (C) pooled specificity, (D) the pooled sensitivity and (E) the pooled diagnostic odds ratio of serum miRNA for diagnosis of EC in Asia.

Table 2 The results of subgroup analysis for the diagnosis of EC.

Subgroup	n	SEN (95% CI)	SPE (95% CI)	PLR (95% CI)	NLR (95% CI)	DOR (95% CI)	AUC	
Country								
China	7	0.74 [0.71–0.77]	0.78 [0.74–0.81]	2.94 [2.30–3.77]	0.35 [0.31–0.41]	9.94 [7.50–13.10]	0.82	
Others	5	0.66 [0.59–0.72]	0.79 [0.71–0.86]	2.98 [1.57–5.65]	0.39 [0.27–0.57]	10.77 [5.22–22.22]	0.84	
Specimen types								
Plasma	5	0.73 [0.69–0.76]	0.80 [0.75–0.84]	3.16 [2.36–4.24]	0.36 [0.30–0.44]	10.28 [7.33–14.40]	0.83	
Serum	3	0.60 [0.51–0.68]	0.85 [0.76–0.91]	4.59 [1.38–15.23]	0.42 [0.26–0.67]	13.05 [3.66–46.61]	0.8	
Blood	2	0.75 [0.68–0.82]	0.75 [0.67–0.82]	2.64 [1.55–4.48]	0.33 [0.25–0.45]	9.30 [5.46–15.80]	N/A	
Saliva	2	0.89 [0.79–0.95]	0.54 [0.37–0.71]	1.88 [1.31–2.71]	0.21 [0.10–0.43]	9.47 [3.49–25.73]	N/A	
Stage								
I–IV	10	0.69 [0.65–0.73]	0.81 [0.77–0.84]	3.13 [2.28–4.30]	0.38 [0.31–0.47]	10.63 [7.70–14.67]	0.83	
Others	2	0.77 [0.73–0.81]	0.70 [0.62–0.77]	2.47 [1.84–3.30]	0.32 [0.25–0.41]	8.97 [5.83–13.80]	N/A	
Gender								
Male	6	0.73 [0.66–0.79]	0.79 [0.71–0.85]	2.73 [1.67–4.46]	0.39 [0.31–0.50]	10.82 [6.09–19.20]	0.83	
Female	6	0.72 [0.62–0.80]	0.77 [0.69–0.84]	2.84 [1.93–4.17]	0.39 [0.29–0.52]	8.97 [5.83–13.80]	0.82	
Age								
>60	7	0.66 [0.61–0.71]	0.83 [0.78–0.87]	3.29 [1.98–5.47]	0.41 [0.32–0.53]	11.47 [7.36–17.88]	0.83	
≤60	3	0.81 [0.73–0.88]	0.68 [0.57–0.78]	2.40 [1.40–4.11]	0.29 [0.19–0.45]	9.10 [4.54–18.21]	0.84	
Total	12	0.72 [0.69–0.75]	0.78 [0.75–0.81]	2.87 [2.28–3.59]	0.36 [0.31–0.43]	10.00 [7.73–12.95]	0.82	
Note:

SEN, Sensitivity; SPE, Specificity; PLR, Positive likelihood ratio; NLR, Negative likelihood ratio; DOR, Diagnostic odds ratio; NA, Not Available; AUC, Area under curve.

Sensitivity analysis

We used sensitivity analysis to better identify how the aggregate impact size varies when an individual study is removed. Practically every study had sufficient weight in this meta-analysis (Fig. 11).

Figure 11 Sensitivity analysis of the included studies.

(A) Goodness-of-fit, (B) bivariate normality, (C) influence analysis, and (D) outlier detection. The numbers within the circles represent their order in Table 1.

Fagan’s nomogram

When we set the pre-test probability at 20%, the post-test chance of positive tests increased to 44%, while the post-test likelihood of negative tests decreased to 7%, according to the Fagan’s nomogram (Fig. 12).

Figure 12 Fagan’s nomogram of miRNA-21 for the diagnosis of EC.

Publication bias

Deeks’ asymmetry test showed a statistically non-significant value (P = 0.99), confirming no evident publication bias (Fig. 13).

Figure 13 Deeks’ funnel plot of miRNA-21 for the diagnosis of EC.

The numbers within the circles represent their order in Table 1.

Discussion

EC is one of the most lethal malignancies in the world with a high mortality rate and correspondingly low survival rate (Kojima et al., 2020). About 50% of the new cases of EC were in Asia due to its high incidence rate and increasing popularity of nitrosamine food in Asian countries (Li et al., 2021; Jakszyn & Gonzalez, 2006; Zhao et al., 2020; Stoner, Wang & Chen, 2007). Accurate diagnosis can be attained by biopsy only when patients have reached advanced stage. Meanwhile, the number of endoscopic and pathological doctors is too small in terms of the large quantities of Asian patients (Zhao, 2012). Thus, novel methods for the early detection of SCC and AC are urgently required to reduce mortality and improve treatment.

As a potential diagnostic biomarker for EC, miR-21 possesses several unique advantages. First, serum miR-21 expression levels are stable and reproducible. One explanation for this remarkable stability is that miRNAs might be chemically modified (e.g., methylation), making them resistant to ribonu-clease activity (Miyoshi et al., 2022). Second, when compared with histopathological examination, serum or plasma miR-21 is characterized by minimal invasion and convenience and easily detected by qRT-PCR. Last but not least, its concentration is specifically correlated with certain types of cancer. MiR-21 modulates the expression of multiple cancer-related target genes such as PTEN, TPM1, and PDCD and significant over-expression of plasma miR-21 was observed even in patients with early-stage EC.

Several studies have reported results of MicroRNAs as diagnostic markers in EC (Wang et al., 2014; Liu et al., 2016; Xie et al., 2013), and we read these studies for reference. Liu et al. (2016) conducted a meta-analysis to assess the overall accuracy of circulating miRNAs in serum/plasma in the diagnosis of ESCC and yielded good diagnostic accuracy, with 79.9% sensitivity, whereas Xie et al. (2013) reported a much lower specificity (51.3%) for miR-451 and 47.4% for miR-144. Meanwhile, age and gender have been shown in other pathologies that could modify the expression of circulating miRNAs. Both of them didn’t explore whether these variables affect the miRNA expression in this pathology. Wang et al. (2014) undertook a systematic analysis to further explore the clinical applicability of miRNAs as novel biomarkers for the diagnosis of EC in Asians. Since only eight publications ranged from 2010 to 2013 were included, it is necessary to perfect the conclusions by further research.

To our knowledge, this is the largest and most thorough examination of miR-21’s diagnostic accuracy in Asian EC patients. We did not limit publication date during the search of the six main databases: Ovid MEDLINE, PsycINFO, PubMed MEDLINE, Embase, Web of Science, and the Cochrane Library, making it unlikely to miss important publications. This quantitative analysis includes 14 studies from 12 articles, with a total of 987 EC patients and 625 healthy controls.

Every study in this meta-analysis had sufficient weight, indicating the stability of this meta-analysis. With regard to subgroup analysis, results revealed that: I–IV stage had better diagnostic values than non-I–IV (DOR, 10.63 vs 8.97); serum had better diagnostic values than other specimen types (DOR, 13.05 > 10.28 > 9.47 > 9.30); male had better diagnostic values than female (DOR, 10.82 vs 8.97); Age >60 had better diagnostic values than Age ≤60 (DOR, 11.47 vs 9.10). Deeks’ asymmetry test showed a statistically non-significant value, confirming no evident publication bias.

Pooled sensitivity, specificity, and AUC revealed satisfactory diagnostic accuracy of this meta-analysis (Liao et al., 2015). Positive and negative likelihood ratios are two extremely stable indicators that can fully reflect the diagnostic value of a screening test (Fuchs et al., 2018). As shown in this meta-analysis, we found a pooled PLR of 2.87 which indicated about three-fold increment in the probability of EC in an Asian with a positive relative to a healthy individual. The pooled NLR of only 0.36, on the other hand, meant that the risk of EC in an Asian with a negative test was 36 percent higher than in a healthy person. In addition, DOR was found to have a strong diagnostic effectiveness in 12 investigations.

Despite our best efforts, there are several limitations to the present study: (1) study is based on published articles, and unpublished studies or on-going were not included; (2) the sample size in our meta-analysis was small and more high-quality clinical studies are needed to be carried out for validation; (3) Ten of the studies in this meta-analysis had obvious differences in time or place. However, there may have few overlapping patients between two studies (with obvious differences in specificity and sensitivity) (Sun et al., 2019; Zhang et al., 2018), and emails sent to the corresponding authors got no response.

Conclusions

Our meta-analysis findings imply that circulating miR-21 has potential diagnostic utility for EC in Asians, with good sensitivity and specificity. However, further high-quality research on the diagnostic function of circulating miR-21 should be undertaken in the future.

Supplemental Information

Supplemental Information 1 The raw data for State.

Click here for additional data file.

Supplemental Information 2 The raw data for Meta-disc.

Click here for additional data file.

Supplemental Information 3 PRISMA checklist.

Click here for additional data file.

Additional Information and Declarations

Competing Interests

Author Contributions

Data Availability

The authors declare that they have no competing interests.

Zheng Han conceived and designed the experiments, performed the experiments, analyzed the data, prepared figures and/or tables, and approved the final draft.

Lingbo Pan performed the experiments, prepared figures and/or tables, and approved the final draft.

Bangjie Lu analyzed the data, prepared figures and/or tables, and approved the final draft.

Huixia Zhu conceived and designed the experiments, authored or reviewed drafts of the article, and approved the final draft.

The following information was supplied regarding data availability:

All the raw data are available in the Supplementary Files.

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
