# Peer review of "MicroRNA-21 as a potential biomarker for detecting esophageal carcinoma in Asian populations: a meta-analysis"

_PeerJ, doi:10.7717/peerj.14048_

## Round 0.1 · original submission · Major Revisions

Please review and attend to the reviewers' concerns.

·

Basic reporting

Abstract

1. The present study proposes the miR-21 as an important tool for the early detection of esophageal carcinoma, however several questions can be address:

• The usefulness of this miR for prognosis and/or diagnosis in different populations and similar work has previously been demonstrated (Salman Guyara 2018 PMID: 30336280, Hossein Aalami et al. 2021 PMID 33434077, Song Gao et al, 2018 PMID: 30420592) Why should the outcome in Asian population be different?
• Given the previous point, how is the substantial difference between previous works and this one?
1. The authors refer in this meta-analysis, to miR-21 indistinctly, however, there are several isoforms of this miRNA, the most important, miR-21-5p and miR-21-3p. Both have demonstrated completely different associations and applications (Báez-Vega et al. 2016 PMID: 27166999, Weijuan Jiao et al. 2017 PMID: 28734015, Xiaofen Sun et al. 2020 PMID: 33455113, Zhikui Gao et al. 2019 PMID: 30979011, Jing Song et al. PMID: 34439276, Yang Xia et al. 2019 PMID: 30134284, Xiaohui Li et al 2018 PMID: 29680210). Therefore, it would be worth differentiating them in this study to find the difference mentioned in point 1 and seek to reinforce the results of this work.

2. The language used is very colloquial, improving the wording would significantly benefit the understanding of the work, an example of this is lines 6 and 7 “Hence, we conducted this meta-analysis, hoping to 7 determine whether miR-21 can be used as a diagnostic marker and assess its diagnostic effect”

3. The conclusion is not clear lines 19-20 "with a medium diagnostic value". miRNAs are expensive and difficult to store and work with, if the advantages they offer are not strong when faced with other current methods, then why the relevance?

4. Keywords misspelling


Introduction

1. The introduction is very superficial, it is not clear why if miR-21 has been associated whit other types of cancers and different populations it is particularly important in this work and this population.

2. The language is colloquial, confusing, and difficult to understand:
- line 25-26: “Everyone knows that Asia, particularly China, Iran, and Japan, have the highest rates of EC in the World”
- line 27-28: “Globally, the incidence of EC is thought to be decreasing. However, according to a recent comprehensive analysis, the global burden from EC will continue to climb”
- line 40-41: “As a result, we did this meta-analysis in order to see if miR-21 can be utilized as a diagnostic marker for EC in Asian people and if so, how effective it is.”

Experimental design

Inclusion criteria

1. Age and gender have been shown in other pathologies that could modify the expression of this miRNA. Could these variables affect the miRNA expression in this pathology?
2. Change: Web of Science
3. Line 66: Data extraction
4. It would be important to add a new subtopic including the statistical analysis that was followed, the heterogeneity between the included studies, the Cochran Q test, and the Higgins I-square statistic, etc. Author mentioned in the abstract, but not in the text.
5. ¿Do the studies reviewed include all types of carcinoma? That is, ¿do they include squamous cell carcinoma, adenocarcinoma, sarcoma, small cell carcinoma, or lymphoma? It would be better to specify this in the inclusion criteria

Validity of the findings

Results

1. Line 85-86 is confusing, “There were twelve studies that matched all of the requirements”
2. The Forest Plot results for sensitivity and specificity of miR-21 should be improved, due the heterogeneity of the results is very high. When reviewing some of the studies used, I consider that it could be because the samples where the extraction and measurement of miR-21 were carried out in different corporal fluids, that is, some are whole blood, saliva, plasma, and serum, it would be advisable to subdivide these samples to obtain better results. Remember that the expression values of this miR could be modified depending on where they are searched.

3. This section should be the most important part of the manuscript and lacks strong, and relevant data, in addition, the manuscript doesn’t present different results from those already expressed in previous texts. (Wenjie Zhu et al, 2014 PMID: 25098165, Huiying Zhang et al, 2021 PMID: 34266326, Salman Guyara 2018 PMID: 30336280, Hossein Aalami et al. 2021 PMID 33434077, Song Gao et al, 2018 PMID: 30420592).

4. The overexpression or low expression of these miRNAs is what makes them useful, so it should be detailed if the elevation or decrease of miR-21 is related as a marker to diagnosis, treatment, etc. (Wenjie Zhu et al, 2014 PMID: 25098165)

5. The authors repeatedly mention the use of PLR and AUC, etc. but there is no clear explanation of the importance of these results. Improving the wording will make better understood of the study.

Discussion

1. The authors make this section very repetitive, using the results already shown previously. It lacks a good sense and order; it doesn’t compare with similar results from other authors that are important for its analysis and discussion.

2. The authors mention that the use of miRNAs is non-invasive, easy to use, and highly stable, but this is not completely true. In addition, in this section, and with that comment the question is reinforced... if the advantages they offer are not strong when faced with other current methods, then why the importance?

3. In line 39 the authors mention "in blood samples, the results have been mixed however throughout the text the reason for “mixed” is not mentioned. In addition, they mention "in blood samples" but the articles included other fluids, which I consider that both points could be addressed and reinforce the discussion section.

Conclusions

1. The language used is very colloquial, it is not a strong conclusion and it not written in a concise and clear way.

Reviewer 2 ·

Basic reporting

1. The introduction section should consider adding some high-quality literature. Paper would benefit from a deeper analysis of previous works in the field.
2. Some references are very old, so it is suggested to replace them with the latest research in related directions. Such as: “Song J, Bai Z, Zhang J, Meng H, Cai J, Deng W, Bi J, Ma X, and Zhang Z. 2013. Serum microRNA-21 levels are related to tumor size in gastric cancer patients but cannot predict prognosis. Oncol Lett 6:1733-1737. 10.3892/ol.2013.1626.”
3. The language of the paper should be improved by the native speakers.

Experimental design

The description of statistical analysis should be provided in detail. For example, Sensitivity analysis? Subgroup analysis? Publication bias?

Validity of the findings

in Discussion, author said “there are several limitations to the present study”, but listed only two limitations.

Reviewer 3 ·

Basic reporting

The manuscript is in general well written. Authors provide good references with sufficient background information. The manuscript is structured professionally and self contained. Below are aspects for further improvements:
1) Line 53 typo: "AND operator: 1"?
2) Table 1: legend explains acronyms used in Table 2, but not Table 1.
3) Line 84-85: written sentence is opposite from what authors meant. Consider modify to: "532 records were left after elimination based on..."
4) Line 97 typo: negative (Fig. 5)?
5) Please double check figure references in manuscript. Figures are wrongly referenced at multiple places in the manuscript: line 99, line 103, line 112...
6) Line 123 typo: healthy pregnancies?
7) Line 125: please check grammar
8) Line 140-141: weird line break
9) Figure 4: Please increase figure resolution
10) Figure 6 typo: Guiying?Sun

Experimental design

This manuscript is an original primary research with a well defined and meaningful research question. However, the description for methods is too brief for replication. For example, it is unclear how to generate Figure 8. In general, please describe what softwares were used at each step? What was the version of the software? What parameters did the authors use to generate the result?

Validity of the findings

The authors did a good job with data analysis and provided underlying data for this study in Table 1. The conclusion could be further strengthened if authors could check:
1) Is there overlapping patients among 12 studies?
2) Is there gender difference in diagnosis using miR21?

---

## Round 0.2 · accepted · Accept

Thank you for the professionalism in replying to each previous concern from reviewers and me.

In my opinion, your rebuttal letter gives an opportune response to previous issues.